

# 1 A Spectrum of Geoscience Communication:
# 2 From Dissemination to Participation

Sam Illingworth[1]
[1]Department of Learning and Teaching Enhancement, Edinburgh Napier University,
Edinburgh, Scotland

## 8 Abstract

This article is a written contribution to accompany the 2023 Katia and Maurice Krafft
Award from the European Geosciences Union. Though a consideration of my own
practice and that of the wider literature, I investigate whether employing creative
approaches can enhance the diversification of geosciences and facilitate broader
engagement in its research and governance. I propose a spectrum for geoscience
communication, spanning from dissemination to participation, and contend that
effective communication demands a creative approach, considering the
requirements of diverse audiences. I offer practical recommendations and tactics for
successful geoscience communication, including audience awareness, transparency,
and engagement with varied communities. This article emphasises the significance
of fostering increased recognition for science communication within geosciences and
promoting wider engagement in its research and governance. It delivers valuable
insights for researchers, educators, communicators, and policymakers interested in
enhancing their communication skills and connecting with diverse audiences in the
geoscience domain.

## 24 1. Introduction

In 2023 I was awarded the Katia and Maurice Krafft Award from the European
Geosciences Union (EGU). This award, named in honour of the volcanologists Katia
and Maurice Krafft (Calderazzo, 1997), recognises researchers who have developed
and implemented innovative and inclusive methods for engaging with and
communicating a geoscience topic or event with a diverse audience. As part of this
award, I was invited to give a lecture at the 2023 EGU General Assembly and to also
provide a written contribution, based on this lecture, to one of the EGU journals.
Given that a large part of my award and subsequent lecture was grounded in the
work that I have done since helping to found *Geoscience Communication* in 2018, it
seemed as though this would be the most appropriate place for such an article.
The purpose of my lecture, and hence this article, it to attempt to explore the
following hypothesis:



"A creative approach can help to diversify the geosciences and enable more
people to engage with its research and governance, from dissemination to
participation."
In attempting such an exploration, I would first like to introduce the concept of a
'spectrum for geoscience communication'.
I have written elsewhere (Illingworth, 2022, Illingworth and Allen, 2020) about the
need for inward-facing and outward-facing science communication. That there is a
need for science to be inwardly communicated to other scientists (via e.g., peer-
reviewed research articles and conference presentations), and a need for science to
be outwardly communicated with non-scientists (e.g., via policy documents, radio
programmes, and collaborative workshops). In developing this argument, I would like
to present this outward-facing side of science communication, and hence geoscience
communication, as existing on a spectrum, with dissemination at one end, and
participation at the other (see Figure 1).

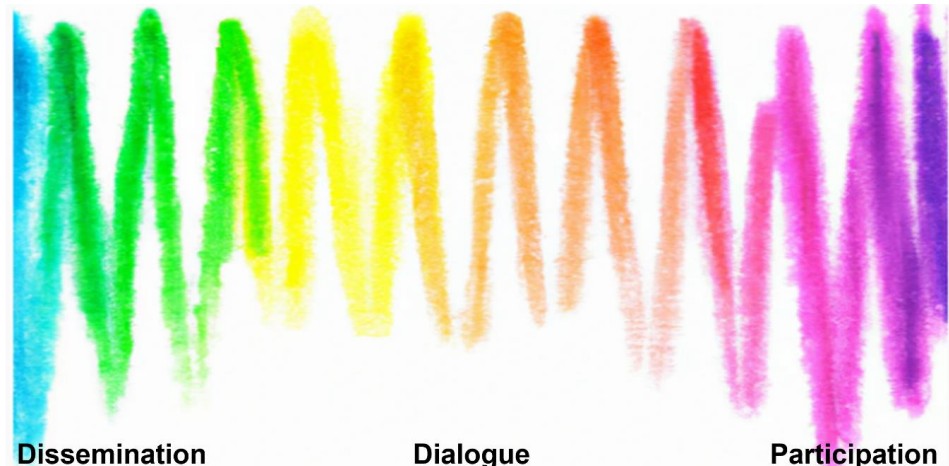

**Dissemination**                **Dialogue**                **Participation**


*Figure 1: The spectrum of geoscience communication, from dissemination to*
*participation (image created using DALL-E with the prompt "the electromagnetic*
*spectrum as a watercolour").*
Although many might consider participation and dialogue to be the ideal approach for
science communication, some goals may be better achieved through dissemination.
For example, science documentaries whilst unidirectional from scientific to non-
scientific publics have been shown to potentially have an impact at a wider societal
level (Dunn et al., 2020). Likewise, providing accurate and easily understandable
information is often a crucial prerequisite for initiating dialogue and with it,
participation (Resnik et al., 2015).
In other words, Fig. 1 is not a hierarchical spectrum, but rather a tool to help identify
the form of a particular geoscience communication initiative. In doing so, it is first
necessary to consider both the aims of the initiative and the needs of the audiences.



For example, if you are interested in developing relationships with local communities
and decision-makers to reduce negative volcanic impacts and uncertainty (Marin et
al., 2020) than you would likely need to engage in some form of dialogue. Similarly, if
you are aim to engage multiple publics to recover old records of sub-daily weather
observations at sea in order to make them useable in current climate models
(Hawkins et al., 2019), then a more participatory approach would be appropriate.
It's crucial to recognise that there isn't a single 'general public'. Instead, multiple
publics exist, each with their unique challenges and possibilities for engagement, as
well as their motivations for engaging (or not) with science (Illingworth and Wake,
2021a). When deciding which public to engage with, it is therefore essential to
carefully consider what and why you want to communicate, as well as the reasons
for interacting with your chosen audience.
In utilising this spectrum for geoscience communication, I also believe that a creative
approach is effective for several reasons. Creative methods simplify complex
concepts by employing techniques such as storytelling, analogies, and visualisation,
making the subject matter more accessible to non-experts (Schäfer and Kieslinger,
2016). They also enhance retention, as entertaining and emotionally engaging
content is often more memorable (Wilkinson and Weitkamp, 2020), and facilitate
dialogue and interaction between geoscientists and non-geoscientists, promoting
collaborative learning experiences (Illingworth, 2020a). Additionally, a creative
approach has been shown to foster interdisciplinary collaboration between
geoscientists and professionals from other disciplines, such as artists, educators,
and communicators, leading to innovative ways of presenting geoscience information
and reaching broader audiences (Illingworth, 2022).
In addressing my hypothesis, I will spend the remainder of this article investigating
the three distinct sections of this spectrum: dissemination, dialogue, and
participation, outlining examples of effective practice for each using creative
methodologies. In doing so I will present an overview of my research into using
poetry and tabletop games as facilitatory media to help disseminate knowledge,
develop dialogue between scientists and non-scientists, and engender participation
amongst diverse publics, including those audiences that have previously been
marginalised by the geosciences.
In addition to my own research, I will also explore how the work that we are doing
with *Geoscience Communication* is supporting others in developing innovative and
effective research and practice in this space, and how this in turn is helping to
provide greater recognition for science communication in the geosciences. In doing
so I hope to outline what makes for effective geoscience communication, and why I
believe that a creative approach is one way in which we might do this.

**2. Dissemination**



Geoscience research can be complex and technical, making it difficult for non-
specialists to understand and appreciate its significance. However, by using poetry
as a means of science communication, geoscientists can convey their research in a
more accessible and engaging way (Young and Kulnieks, 2022). Poetry can help to
simplify complex scientific concepts and make them more relatable to a wider
audience (Wardle and Illingworth, 2022). For example, a poem about the impact of
climate change on glaciers could use vivid imagery and metaphors to convey the
beauty and fragility of these natural wonders, while also highlighting the urgent need
for action to address climate change (Illingworth, 2016).
In addition to making geoscience research more accessible, poetry can also help to
create emotional connections with readers or listeners. By evoking emotions such as
wonder, awe, or concern, poetry can inspire people to care about geoscience issues
and take action to address them. This is particularly important when it comes to
issues such as the climate crisis or natural disasters, which can often feel
overwhelming or abstract (Illingworth, 2020b). Poetry can help to humanise these
issues and make them more tangible (Anabaraonye et al., 2018).
Like poetry, tabletop games are effective at disseminating geoscientific research to a
non-specialist audience for a variety of reasons. In using the phrase tabletop game, I
mean any non-digital game that can be played on a table (e.g., card, dice, and board
games). When it comes to geoscience communication, the advantages of tabletop
games, compared to their digital alternatives, may encompass factors such as cost
(regarding development, technology, and resources), adaptability (allowing players
or educators to effortlessly modify game parameters to align with their educational
objectives, time, and space constraints), and most notably, the manner of
engagement, which typically involves direct player interaction (Illingworth and Wake,
26  2019).

Tabletop games inherently engage participants through their interactive and
entertaining nature, making them more likely to retain information and maintain
interest in the topic (Pfirman et al., 2021). Such games are also a fantastic medium
for simplifying complex concepts; they have the capacity to break down unfamiliar
geoscientific ideas into more manageable elements (Fjællingsdal and Klöckner,
2020), making them accessible and understandable to non-specialists (Locritani et
al., 2020). Finally, tabletop games encourage active learning, as players must apply
their knowledge and problem-solving skills to progress; this hands-on approach can
promote a deeper understanding and retention of geoscientific concepts.
Other creative media that have proven to be effective at disseminating geoscientific
research to non-specialist audiences include music (Menghini et al., 2020), comics
(Wings et al., 2022), and even letter writing (Stiller-Reeve et al., 2023). Likewise,
despite my earlier (playful) claim that tabletop games are more effective than digital
games, there are many examples of digital games being used as an impactful tool
for dissemination. This has perhaps proven to be most successful when researchers
have used well-known, video game franchises such as Minecraft (Rader et al.,



2021), Monster Hunter (McGowan and Scarlett, 2021), Pokémon (McGowan and
Alcott, 2022), and Zelda (Hut et al., 2019) to explore how the geosciences are
represented (or not) in these game worlds.
**3. Dialogue**
Whilst poetry and tabletop games are effective media for disseminating geoscientific
research from scientists to non-scientists, their real strengths lie in the capacity to
facilitate dialogue between these publics.
To genuinely advance scientific research and discourse, it is essential to address our
social responsibility as scientists and make science accessible to everyone, rather
than an exclusive privilege for a select few. Engaging diverse publics in a genuine
two-way conversation about our research, its relevance to them, and the potential
contributions they can make to new knowledge is crucial. By not establishing this
dialogue, we miss the opportunity to benefit from the expertise of the publics we aim
to communicate with. These publics, although not scientists, possess expertise in
various aspects of their personal and professional lives. By seeking their opinions
and identifying ways to benefit from their knowledge, we (as geoscientists) can
therefore enhance our own understanding and knowledge.
One of the main challenges in creating such two-way conversation is the idea that
geoscientists are experts while others are not. This can make people feel less
important and less likely to share their thoughts, even though they might have
valuable insights about a topic and how it affects society. These obstacles, known as
'hierarchies of intellect' (Illingworth and Jack, 2018), emerge when people are urged
to discuss a subject where one party (i.e., the geoscientist) is perceived as an
expert, while the other (i.e., the other publics) is not. Such hierarchies hinder
effective dialogue and can lead to marginalising audiences, discouraging them from
sharing their knowledge and experiences. Yet these insights might be crucial for a
better understanding of specific research findings and their potential implications on
the broader society.
One way to break down these barriers is by writing and sharing poetry together in a
friendly and supportive setting. This helps everyone feel equal and allows for a true
exchange of ideas between different groups, each with their own knowledge and
experiences. Collaborative poetry sessions are successful in creating dialogue for
three reasons: they show the public that their expertise is valued, they allow
scientists to connect with people on an emotional level, and they create a sense of
shared vulnerability (Illingworth, 2020a).
These collaborative poetry writing sessions are especially effective when engaging
with audiences who have traditionally been under-served or marginalised by the
geosciences. For example, my own work has shown how poetry can help to engage
potentially vulnerable audiences with both the climate crisis (Illingworth et al., 2018)



and environmental change (Illingworth and Jack, 2018) more broadly in a supportive,
constructive, and safe environment. Similarly, other studies have shown how poetry
can be used to develop dialogue between geoscientists and non-scientists on topics
ranging from soil (Maria and Arnalds, 2018) to the conservation of natural heritage
(Nesci and Valentini, 2020).
Similarly, tabletop games are a proven way of developing these two-way dialogues,
mostly because of something that is referred to in game studies parlance as 'the
magic circle' (Stenros, 2014). This circle refers to the imaginary boundary that
separates the game world from reality. Within this circle, players engage in activities
governed by specific rules and structures, suspending real-world norms and
embracing the game's alternate reality. This suspension allows us to move beyond
any hierarchies that may exist outside the gaming context, enabling interactions that
might not be possible otherwise (Illingworth and Wake, 2021a). For instance, in the
board game Monopoly, it is acceptable (if not essential) behaviour to try and
bankrupt your fellow players by levying rental income on multiple properties,
behaviour that (one would hope) is viewed as being morally repugnant away from
the gaming table. Agreeing to abide by a set of arbitrary and sometimes restrictive
rules can help create a secure environment for fostering new interactions and
learning. Doing so helps to break, or at least temporarily suspend, any hierarchies of
intellect, allowing for more inclusive engagement and rich dialogues to emerge.
One example of such a game that does this from a geoscientific point of view is
*Keep Cool*, a climate negotiation game in which players assume the roles of
countries or nations, each with distinct economic interests, objectives, and
capabilities (Fjællingsdal and Klöckner, 2020). The actions players take to achieve
their goals also generate greenhouse gases, and everyone loses if the global
temperature rises too much (Fennewald and Kievit-Kylar, 2013). Each round, players
must decide whether to implement climate protection measures that benefit all or act
in their self-interest to reach their goals more quickly. The first player to achieve their
goal wins, but a total lack of cooperation among players can lead to global
environmental collapse. This game creates a neutral environment where scientists
and non-scientists can interact on equal footing, breaking down barriers and
enabling open dialogue. Similarly, by taking on the roles of different countries with
varying interests, players gain insight into the diverse perspectives and challenges
faced in real-world climate negotiations, fostering empathy and understanding
between scientists and non-scientists.
Likewise, when we designed our 'Global Warming' expansion for the popular
tabletop game *Catan®* (Illingworth and Wake, 2019), we wanted to create a game
(or in this case a modification for an existing game) that enabled geoscientific and
non-geoscientific publics to explore the consequences of individual action and the
extent to which mitigating the negative effects of global warming requires a collective
response.





During the game's playtesting, feedback from various playtesters suggested that the game mechanics, rather than any related story, effectively and elegantly fostered dialogue on a specific subject, such as global warming. We also concluded that to develop a tabletop game for effective dialogue, it is essential to consider the game's accessibility, players' game literacy, the peer review of scientific content, and the degree to which the metagame (i.e., discussions occurring around and beyond the game) is facilitated.

As with 'Dissemination', many other creative forms of geoscience communication have also been uses to foster effective dialogue between geoscientists and non-geoscientists. Such initiatives have included films (Archer, 2020), sculptural work (Lancaster and Waldron, 2020), and printmaking (Macklin and Macklin, 2019). What arguably marks these initiatives out as being especially effective is that they have led to actionable dialogue for the publics involved, rather than just the creation of another 'talking shop' for researchers to share the 'brilliance' of their geoscientific findings.

## 4. Participation

There are two phrases that often get bandied around in public engagement and science communication parlance when it comes to participation: citizen science and co-creation.

Citizen science projects in geosciences, such as those geared towards disaster risk reduction (Hicks et al., 2019), have the potential to both benefit multiple publics and also utilise the lived experience and expertise of non-geoscientists in a tangible and actionable manner. However, concerns arise regarding the potential exploitation of participants as free labour, with scientists reaping the benefits and recognition (Strasser et al., 2019). To address this, it is essential to actively involve participants and acknowledge their contributions, ensuring they are not treated as second-class citizens. Embracing social media and communication platforms can further expand engagement in citizen science projects while promoting fair recognition for all involved (Liberatore et al., 2018). Similarly, creative media such as art and poetry provide a powerful medium through which to challenge and address some of these potential inequities (see e.g. Bauman and Briggs, 2003, Torre and Fine, 2011).

Another issue with citizen science is that some form of training is often essential. Simpler tasks demand minimal training, while more complex ones require extensive instruction. To encourage participation, most projects aim for low training requirements. Nonetheless, adequate training is crucial to maintain data quality. Again, this is where creative methodologies can really help to contribute to the field, with music (L. Oliver et al., 2021) and games (Strobl et al., 2020) both having been shown to be effective (and fun!) ways of providing training in an equitable and effective manner.



Similarly, co-creation is a participation phrase that is often used, yet perhaps with
more fervour than is strictly true or necessary. In true co-creation, collaborations
should start early, involving all participants from the beginning to maximise skill and
expertise benefits (Illingworth, 2022). Including all collaborators in formulating
research questions and aims promotes trust, teamwork, and fosters innovative ideas
enriching the experience for everyone.
A creative example of a genuinely co-creative process is the poetry and art journal
that I help to curate. *Consilience* (https://www.consilience-journal.com/) is the world's
first peer-reviewed science and poetry journal, publishing themed poems and
artwork by creatives from all backgrounds. The journal provides support to develop
the craft and identity of contributors, using a peer review system like scientific
journals. *Consilience* is run by over 80 global volunteers and has around 8,000
monthly readers. The journal was created to help develop the work of others in the
field, transcending individual limitations. Early collaborators defined the journal's
purpose, framework, and submission process.
*Consilience* is a good example of an interdisciplinary collaboration between
scientists, poets, and other creatives, where the co-creation began at the very start
of the project, and through which multiple voices were both present and platformed.
However, whilst the journal is clearly doing good work in helping to diversify the ways
in which science is interrogated and communicated, it is not engaged with the
creation of geoscientific research itself (at least not directly). This is where tabletop
games come in.
The process of designing tabletop games offers an immersive approach to co-
creation in the geosciences, the reason being that designing and playtesting games
is a genuinely collaborative method that involves listening to several different voices,
and then reflecting and acting on these suggestions for input and development.
In 2018, my colleague Paul Wake and I collaborated with the climate charity Possible
to develop workshops exploring heat decarbonisation and the UK's transition to a
zero-carbon economy (Rydge et al., 2018). Utilising games as icebreakers and tools
to generate dialogue, we engaged multiple publics including climate activists,
policymakers, educators, journalists, students, researchers, and industry
professionals. These workshops were designed to gather knowledge from a variety
of communities who all had an interest and expertise in the subject. This knowledge
was collected via participant observation and written responses to questions, which
were then used to create the framework for a card game.
Following an initial design phase, the card game was then playtested with other
members of the same (and similar) communities, with their feedback used to
improve the game in terms of both its narrative and mechanics. The final game
*Carbon City Zero* involved players taking on the role of city mayors and competing
against one another to become the world's first zero carbon city (Germaine, 2022).
The game was made available to download as a free print and play, and a physical



copy of the game was also successfully launched on the crowd-funding platform
Kickstarter.
Following the release of *Carbon City Zero*, further members of the various
communities that had been involved in the research project got in touch with their
own feedback. Most of this feedback was centred around one key issue: why was
the game competitive when for a truly zero carbon world, cities should be working
collaboratively. As a result of this feedback, a second edition of the game was
collaboratively developed and released as *Carbon City Zero: World Edition*
(Illingworth and Wake, 2021b). In this version of the game, players had to work
collaboratively to reduce the carbon level of a single city to zero within a strict time
limit. Players then either collaboratively won or lost together. As game designers and
researchers, we found this to be a great example of why it is important to really listen
to the needs of the various publics you engage with, rather than just assume what
they want.
Overall, this project successfully involved diverse communities, valued their opinions,
and used their expertise to improve the game. Conversely, there were areas for
improvement. Workshop attendees generally shared similar views on a zero-carbon
future, so including dissenting or differently informed voices could have highlighted
more barriers to reducing carbon emissions and fostering dialogue on the topic.
From the feedback that we received following the release of the game, we know that
it has been used as a tool for enacting actual change, e.g., in townhall planning
meetings and grant applications for similar games-based geoscientific research.
However, there are even more effective examples from across *Geoscience*
*Communication* that have used creative methodologies to develop co-creative
partnerships between geoscientists and other publics. This includes using
storytelling to co-create interventions addressing the climate crisis (Woodley et al.,
2022), using science theatre to debunk scientific mistruths (França et al., 2021), and
even a metanalysis of creative practice as a tool to build resilience to natural hazards
in the Global South (Van Loon et al., 2020).

**5. Conclusions**
At the outset of this article, I aimed to investigate the following hypothesis:
"A creative approach can help to diversify the geosciences and enable more
people to engage with its research and governance, from dissemination to
participation."
By providing examples from my own research and practice, alongside other peer-
reviewed and highly impactful examples from the wider literature, I have
demonstrated the potential of creative approaches in geoscience communication.
However, it is important to acknowledge that creative approaches may not always be
feasible or appropriate for every situation. For instance, in cases where conveying



highly technical information is required, an alternative approach might be better
suited to ensure accuracy and clarity. Additionally, certain creative methods might
not resonate with all audience members, so it is essential to consider a wide range of
strategies to maximise engagement.
To address these limitations and develop effective communication strategies with
various publics, here are five recommendations for geoscientists to consider when
looking to develop their own effective geoscience communication strategies:
1. Know your audience. Before communicating any scientific information, it is
important to understand who your audience is and what their interests and
needs are. This will help you tailor your message and delivery to be more
effective. And remember, there is no such thing as the 'general public'.
2. Be adaptable. Recognise that different situations and audiences may require
different communication approaches. Be prepared to adjust your strategy as
needed to best engage your audience. Use the spectrum of geoscience
communication (Fig. 1) to determine the most appropriate method to achieve
your aim with your intended audience.
3. Be creative. Embrace creative methodologies when appropriate to make your
communication more engaging and relatable. This may include poetry,
storytelling, art, games, or other interactive methods.
4. Be transparent. When communicating scientific information, it is important to
be transparent about any uncertainties or limitations in the data or research.
This helps build trust with your audience and promotes open dialogue.
5. Engage with diverse communities. To promote greater recognition for science
communication in the geosciences, it is important to engage with diverse
communities and promote inclusivity in all aspects of research and practice.
By following these recommendations, geoscientists can develop effective
communication strategies that engage diverse audiences and promote greater
recognition for science communication in the geosciences. Embracing creativity and
inclusivity will not only enhance the field of geoscience communication but also help
address global challenges by fostering collaboration and understanding across
disciplines and communities.
**Competing interests**
Sam Illingworth is the chief executive editor of *Geoscience Communication*.

**Ethical Statement**

As the author of this article, I have made every effort to ensure that the research and
practices discussed in this manuscript adhere to the highest ethical standards. All
studies and projects mentioned were conducted in accordance with relevant





institutional and national guidelines, with the necessary approvals and informed
consent from participants when applicable.
I have taken care to provide accurate, balanced, and transparent information, as well
as acknowledging the limitations and challenges of the methods and approaches
discussed. I have also been conscientious about giving proper credit to the work of
other researchers and creatives, with appropriate citations and acknowledgments.
I have no conflicts of interest to declare, financial or otherwise, and have conducted
my research and communication activities with integrity, impartiality, and in the
interest of promoting greater understanding, inclusivity, and collaboration within the
field of geoscience communication.
**Acknowledgements**
I would like to thank all the colleagues and community members who have made my
work in geoscience communication possible. Special thanks also to Mathew Stiller-
Reeve for providing the nomination for my Katia and Maurice Krafft award and to
Louise Arnal, Caitlyn Hall, Rolf Hut, Roxy Koll, and Chris Skinner for providing letters
of support. Thank you also to DALL-E (a deep learning model developed by OpenAI
to generate digital images) for helping me to produce Figure 1 in this article.

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

Case for Climate Change Curricula through Poetic Inquiry that Involves Storytelling and
Walking the Land. *Justice and Equity in Climate Change Education.* Routledge.
