# Peer review of "A Spectrum of Geoscience Communication: From Dissemination to Participation"

_EGUsphere, 2023_

## Community Comment (CC1)

Comments on Illingworth GC ms, 2023-08, by crookall

First, again congrats on your award. I thoroughly enjoyed you talk at the EGU.

My comments below may appear a little harsh (apologies), but my aim is to help you improve an already excellent article. Overall, I found your article extremely interesting, but also a bit frustrating because I felt that it was still a little green, in the sense that some things could do with tightening up and exploring in more depth. I would also provide a table or two summarizing (maybe contrasting) various methods. I hope that my comments are useful. Please feel free to ignore anything with which you do not agree.

| p & line | Comments |
|---|---|
| 2.1 | About your hypothesis:

a. I wonder if this kind of hypothesis is necessary or warranted. As written now, it contains an assumption (hypothesis) of cause and two effects.
b. Could it be restated as something like?: Thoughts on and pathways to progress in using and expanding creative methods to enable more …
c. I realize that you probably need some form of hypothesis. However, the one that you have simply cannot be 'proven', at least not in scientific terms. To be strict, one would need a null hypothesis and then determine probabilities as to how far creativity cab help greater diversity and engagement.
d. I would like to see the rest of your the various parts of your article more explicitly tied to the hypothesis. |
| | I am not sure if you wish to have a section on materials – which are essentially the literature and your reasoning. |
| Fig1 | I like the spectrum, but I am not sure if the image conveys the message. That is of course very subjective. Maybe just the standard visual spectrum would be stronger? In any case, I wonder how your three types or categories constitute a spectrum. Would you not need several more, each one merging into the next? I wonder if just three categories would make things clearer? Maybe with a table, with basic indications of each category (on the y axis), such as description, main media, type of audience, ease of implementation, etc., each with one to four stars, for example. |
| 2.19 | Lots of films are also lectures, discussions, debates, etc. |
| 3.3 | Could dialogue be equated with social interaction? |
| 3.7 | I personally ban all abbreviations from my writing. My subjective reaction is that they make the writing for an academic journal look a little unkempt. |

| | |
|---|---|
| 3.13 | Creative.  I would welcome a paragraph, or more!, on what this is.  It is an essential tenet of your article and I my view warrants some clarity.  My bible for this Koestler's wonderful book, *The Act of Creation*. |
| 3.32 | Marginalised.  This seems to be a rather strong term, almost as if goesci had wilfully pushed some people away. |
| 4.1 | I will not comment on poetry as I have zero qualification there.  However, having worked with games for some 45 years, I think that I can offer some valid throughts. |
| 4.17 | In simulation gaming, we tend to use the term manual game – or manual simulation/game.  You might wish to cite other work on the advantage of manual games, compared with their computerized counterparts – but this is not necessary – in any case, none come to mind :-) |
| 4.29 | I personally, would not use the word fantastic.  One gripe that I have against much gaming is that it uses too much hype.  Some gamers exaggerate the wonderful/fantastic effect of games without really knowing whether it is well founded. |
| 4.27 | It might be worth developing (explaining a little more) terms such as engagement (see, eg, work by Whitton), active learning (a term that is banded about more than it probably should).  The entertainment value of games is in my view over worked, and does not usually correspond to players' experience. |
| 4.35 | "deeper understanding and retention of geoscientific concept".  A major area too is skills.  Maybe reword something like:  deeper understanding of geo processes, greater retention of geoscientific concepts and hone a wide range of skills (such as critical thinking, research methods, listening, etc |
| 4.39-5.4 | Put this para higher up, where you first talk about simulation/games – it sounds like is was tacked on as an afterthought.  Mention games in general, and then say that you have selected manual games – for the reasons that you give. |
| 5.12 | "two-way conversation" – Dick Duke, one of our pioneering gamers, talks about multilogue.  Here, I wonder if the term interaction would also help?  Interaction among people, with also with geo concepts and geo phenonmena. |
| 5.19 | idea.  Maybe it would be more accurate to qualify it as presumption or even prejudice? |
| 5.25 | Would it make sense to have a ^para in the notion of expert – what do different audiences mean by this term?  Is it the term that we should use? |
| 5.27 | Also:  many insights are contained in the 'non'sci lit, eg, blogs, magazines, online discussions (eg, LinkedIn), etc --  how far do geosci people read those, and crucially how much do cite and quote them?  If lay people were to be cited, that would, in my view, be a good pull for them to come closer to geo sci.  People are |

| | |
|---|---|
| | always pleased when they are cited, and it can endear some who feel that goe scientists are aloof, when they are not. |
| 6.6 | "games are a proven way of developing these two-way dialogues" – sorry, no, they are not proven, not in the scientific sense and not in lay sense of the term.  This is even more so as much missed learning, dialogue, etc are simply squandered by failing to debrief games fully and properly.  [I will send you a publication on that.] |
| 6.8 | Magic circle.  Quite a few gamers have trouble with this term – and I do too.  You yourself describe it as an "imaginary boundary".  If it is imaginary, then it is not real.  If it is a boundary, then it is neither the game nor reality.  I accept "suspension of disbelief" (SD), and that has been mentioned many years ago as an element in a simulation/game.  However, SD is not a result of any so-called magic circle.  Ot is the capacity of the human mind to put aside some things, to be able to concentrate on others.  [My hunch is it may well be at the basis of denial, which is screwing up life on this planet.] |
| 6.11 | I think that a game is not an 'alternate reality' – it is a reality in its own right, and contains many of the elements of non-game reality (otherwise players would be able to operate). |
| 6.17 | Arbitrary.  If rules in games were arbitrary, we would have a riot on our hands.  Unless players can see that a rule serves some purpose, then are unlikely either to follow it or to earn from it.  Generally, rules in simulation/games are of two types: 1. The represent similar rules in the referent (the real world), 2. They make the game work in certain ways. |
| 6.30 | Neutral environment (NE).  They may constitute a NE if they have been designed that way.  However, many games carry an underlying or implicit political, social or prescriptive message (eg, you should not waste food, you should eat less meat).  Indeed monopoly was originally designed by someone who wanted people to know about how ruthless and greedy landlords are, and it is often used (and of course debriefed) in sociology courses to teach about the rich/poor divide. |
| 7.5 | I really like the odea of game literacy.  In my training courses, I emphasize that it is often necessary to ease novice game participants into this new mode, so that they can become a little gale literate before can expect then to learn from the game. |
| 7.6 | By metagame, do you mean debriefing?  In any case, in my view an article about educational or social games that does not include de debriefing is incomplete – and is doing a disservice to the use of games. |
| 7.9 | uses should probably be used |
| 7.13 | I wonder whether 'actionable dialogue' might benefit from a short explanation with some concrete examples? |

| | |
|---|---|
| 7.22 & 23 | I do not think that you need both 'both' and also 'also' |
| 7.33 | Yes, training is so essential. |
| 7.37 | 'really' not needed – or maybe 'help greatly' |
| 8.2 | What is 'true co-creation' (CC)? as opposed to false CC or unreal CC? |
| 8.23 | Debriefing must be integral to the simulation/game design process. |
| 8.39 | *Carbon City Zero* seems to have several websites, some seemingly unrelated to your game. Would be good to provide a URL. |
| 9.15 | You might be interested in Companion Modeling (ComMod), where, thought an iterative process (often over several months) participants help to build the underlying model of the game, and then participate in a role-play game, followed by another round, until the participants themselves declare that the game represents their world. |
| 9.18 | "demonstrated the potential of creative approaches" – this is a key phrase, and should probably be the central idea in your initial hypothesis |
| 10.6 | May I suggest that you also ad something like fully debrief simulation/games? |

---

## Author Response (AR1)

A point-by-point response to the reviews including a list of all relevant changes made in the manuscript is on the online discussion forum.

---

## Author Response (AR2)

Solmaz,

Thank you for these very helpful suggestions, all of which I address below (your comments in bold).

**Page 1, line 9: I suggest rewording the first sentence to emphasize that this is a review article: 'This review article is a written contribution ...'**

Done.

**Page 1, line 10: Correct typo: through**

Done.

**Page 1, last paragraph (starting with lines 37-28): Here again I suggest rewording to include reviewing and summarizing/highlighting previous work on using creative approaches in geoscience communication as one of the aims of this article.**

Done.

**Pag 2, line 9: Revise sentence: e.g., via (to keep it consistent with how you do this throughout the manuscript)**

Done.

**Page 3, line 14: Revise to state: It is, is not (more formal language)**

Done.

**Page 3, line 20: Consider deleting 'I also believe that a'.**

Done.

**Page 4, line 6: Replace 'believe' with 'propose'.**

Done.

**Page 4, line 22: Replace 'natural disasters' (a misleading term) with just 'disasters'.**

Done.

**Page 4, line 24-25: Consider deleting ' Whilst I do not consider myself to be the world's most accomplished poet, I offer...' and start the sentence with 'the following poem is an example...'**

Done.

**Page 7, line 11: If you write that digital games are equally effective, then there is no reason to say they are not effective. It may confuse the reader as it is written now. I suggest starting the sentence with 'There are many examples of digital games being used as an impactful (and equally effective) tool for dissemination.'**

Done.

**Page 8, line 17: Revise '...are crucial.'**

Done.

**Page 10, line 16: Could you provide a copy of the survey form? It can be included as Supplemental Information.**

This work appears in another published study, and so because of IP issues, I have inserted the following text:

In some cases, paper copies were provided, with the authors manually inputting playtester responses into Google Forms (see Illingworth and Wake, 2019 for a copy of the survey form that was used in this study).

**Please provide the links to Carbon City Zero and Catan®: Global Warming games either in the text or in references.**

I already have references to the games, for Catan®: Global Warming this is Illingworth and Wake, 2019 and for *Carbon City Zero* this is Germaine, 2022. I prefer not to use hyperlinks in case they get deleted in the future, but if needed can add in the following:

*Carbon City Zero:* https://boardgamegeek.com/boardgame/288179/carbon-city-zero

*Catan®: Global Warming*: https://boardgamegeek.com/boardgame/305516/catan-scenarios-global-warming

I have also made some very minor additional changes to the manuscript; mainly removing my use of the word 'important'!

Thank you again for all of your help and guidance in the editorial process, which I believe has really strengthened the work.

Sam